# Evaluating Adversarial Defense in the Era of Large Language Models

## Abstract

Large language models (LLMs) have demonstrated superior performance in many natural language processing tasks. Existing works have shown that LLMs are not robust to adversarial attacks, questioning the applicability of these models in scenarios with safety concerns. However, one key aspect that has been overlooked is evaluating and developing defense mechanisms against adversarial attacks. In this work, we systematically study how LLMs react to different adversarial defense strategies. We also propose defenses tailored for LLMs that can significantly improve their robustness: First, we develop prompting methods to alert the LLM about potential adversarial contents; Second, we use neural models such as the LLM itself for typo correction; Third, we propose an effective fine-tuning scheme to improve robustness against corrupted inputs. Extensive experiments are conducted to evaluate the adversarial defense approaches. We show that by using the proposed defenses, robustness of LLMs can increase by up to 20%. Our code will be publicly available.

## 1 Introduction

Large language models (LLMs) such as the decoder-only GPT family (Chowdhery et al., 2022; OpenAI, 2023; Touvron et al., 2023a;b) and the encoder-decoder T5 family (Raffel et al., 2020; Chung et al., 2022) have demonstrated superior performance in various tasks, including natural language understanding (Chung et al., 2022), dialogue generation (Bubeck et al., 2023), logical reasoning (Bang et al., 2023), and even solving mathematical problems (Frieder et al., 2023). These models contain billions of parameters, and their emergent abilities (Wei et al., 2022) facilitate effective zero-shot learning. That is, LLMs are adept in performing tasks in different fields (Choi et al., 2023) using proper prompts without prior exposure to task-specific data.

LLMs are double-edged swords. Despite the successful applications, these models are not *robust*. Figure 1 demonstrates behavior of ChatGPT under input typos. In the example, ChatGPT is asked whether the word "pretty" is of positive sentiment. Without any typos, the LLM correctly answers the question. However, when there is a typo in the input, i.e., "prettye" instead of "pretty", Chat-GPT draws an opposite conclusion. The lack of robustness issue exists beyond typos in input data. For example, prompts are also subject to attack: we can inject backdoor triggers (Xu et al., 2022) or adversarial demonstrations (Wang et al., 2023b) into prompts to trick LLMs in drawing wrong conclusions. These findings raise a serious safety concern: can LLMs be reliably used?

We consider plausible scenarios in real-world applications: **input typos**. In practice, user inputs are often noisy and contain typos, which is undesirable for applications such as dense retrieval and search. For example, Zhuang & Zuccon (2021; 2022) show that retrieval recall can decrease by more than 20% when dealing with input typos. The study of robustness plays a crucial role in mitigating such a performance drop. In this work, we simulate input typos via character-level adversarial attacks (see Figure 1 as an example), and we term the attacks **adversarial typos**. The adversarial typos represent the *worst case input typos*, since they are intentionally created to fool the LLMs. We remark that there are other types of adversarial inputs, such as distracting samples (Wang et al., 2021) and automatically computed adversarial samples (Goodfellow et al., 2015). However, most of these adversarial attacks are not human interpretable and implausible in actual applications.

In this work, we focus on evaluating and developing **defense mechanisms** against adversarial typos, which is a key aspect overlooked by existing works on benchmarking robustness of LLMs (Chen

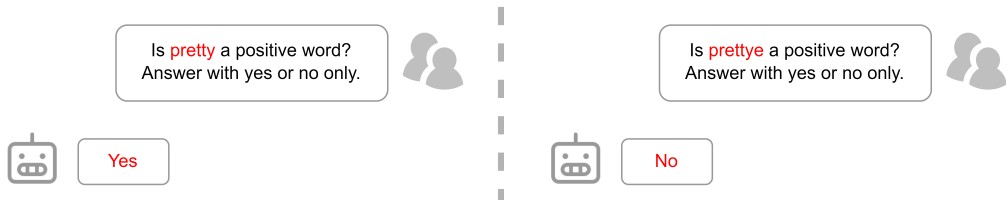

Figure 1: Behavior of ChatGPT under user input typos.

et al., 2023; Wang et al., 2023a). In practice, somehow surprisingly, we observe that many existing adversarial defense methods do not work well for LLMs (e.g., the rule-based defenses in Table 3 in the experiments). Therefore, adversarial defenses that are tailored for LLMs are needed.

Adversarial defense methods fall into two categories: **black-box defenses** and **white-box defenses**. The former describes the scenario where we do not have access to the model weights (e.g, GPT-4); while in the latter case, we have full access to the model (e.g., Llama).

For black-box defenses, we treat defense models as pre-processors: given a potentially corrupted input, we first run the pre-processor, and then feed the processed results to the LLM. We evaluate and develop several categories of methods: 1) *Rule-based defense* methods employ clustering algorithms to learn mapping rules, which are used to convert potential typos (Jones et al., 2020). For example, if the word "hallo" is in the cluster centered around the word "hello", then all the "hallo" in the inputs will be converted to "hello". 2) *Prefix defense* is a simple strategy where we modify prompts to alert LLMs about potentially adversarial contents. 3) *Neural network defense* methods are models trained to denoise typos. For example, Pruthi et al. (2019) train a sequence-to-sequence model to spell-check inputs. 4) *Self defense* methods are similar to the neural network defense, except that the same LLM is used for both typo correction and inference.

We also evaluate white-box defense approaches. With the development of LLMs, more models are becoming fully available (e.g., Llama). This paves the way for designing stronger defense techniques by utilizing the model weights. Specifically, we fine-tune LLMs on both clean data and corrupted (adversarial) data. The intuition is that once exposed to adversarial typos, models should yield better robustness against them.

We evaluate effectiveness of defense approaches on two families of LLMs: decoder-only models such as Llama, and encoder-decoder models such as Flan (Chung et al., 2022). There has long been debates about the suitable structures of Transformer-based models (Vaswani et al., 2017), even before the era of LLMs. For example, the discussion about BERT vs. GPT vs. T5 continued until the advancement of ChatGPT. In this work, we systematically investigate how different LLMs behave when facing adversarial typos and defense mechanisms.

In summary, in this work we consider input typos, a practical but under-explored scenario that hinders performance of LLMs. We systematically study how LLMs react to different defense approaches against adversarial typos. We summarize our contributions and findings as follows:

⬦ We can adopt the LLM itself or train another smaller model to denoise the typos. This black-box defense approach improves robustness of all LLMs.

⬦ We can modify the prompt to alert the LLM in paying attention to potential adversarial contents. Such a black-box defense strategy improves robustness when the LLM is adept in following human instructions, e.g., Llama2-Chat.

⬦ For white-box defenses, we can fine-tune the LLM on both clean and corrupted data. This simple strategy is extremely effective in improving model robustness.

## 2 BACKGROUND

### 2.1 LARGE LANGUAGE MODELS

Large language models have become the de facto standard solution for many natural language processing tasks. These models are a step forward from smaller language models such as BERT (Devlin

et al., 2019), RoBERTa (Liu et al., 2019), DeBERTa (He et al., 2021b;a) and T5 (Raffel et al., 2020). LLMs have extremely large number of parameters compared with their predecessors. For example, DeBERTa-xxl, the largest publicly available encoder-only model in its series, contains about 1.3B parameters, whereas LLMs rarely have less than 10B parameters.

The debate about the best architecture of Transformer-based models (Vaswani et al., 2017) continues in the era of LLMs. One commonly used structure is the decoder-only Transformer. The most famous example is the GPT series: starting from GPT (Radford et al., 2018), GPT-2 Radford et al. (2019), GPT-3 (Brown et al., 2020) and InstructGPT (Ouyang et al., 2022), the series has now progressed to the evolutionary GPT-4 (OpenAI, 2023). Other examples include Llama (Touvron et al., 2023a;b), PaLM (Chowdhery et al., 2022), OPT (Zhang et al., 2022) and BLOOM (Scao et al., 2022). Another widely applied structure is the encoder-decoder Transformer, evolving from T5 (Raffel et al., 2020) and BART (Lewis et al., 2020) to the Flan family (Chung et al., 2022) including Flan-T5 and Flan-ul2. Many existing works on benchmarking robustness of LLMs focus on the decoder-only GPT family (Wang et al., 2023a; Chen et al., 2023). We bridge this gap by evaluating both decoder-only and encoder-decoder models.

## 2.2 ADVERSARIAL ATTACKS

In this work, we focus on models' robustness to input typos, which is a practical scenario that hinders model performance in real-world applications such as dense retrieval and search (Zhuang & Zuccon, 2021; 2022). Specifically, we use character-level adversarial attacks to simulate the *worse case input typos*. That is, the modifications to the input (i.e., the typos) are intentionally designed so that the LLM draws wrong conclusions. Existing works have designed other character level (Belinkov & Bisk, 2018; Gao et al., 2018; Eger et al., 2019), word level (Papernot et al., 2016; Zhao et al., 2018; Alzantot et al., 2018) and sentence level (Iyyer et al., 2018) attacks to mislead the models.

We note that in parallel to typos in the input data, LLMs are also not robust to adversarially constructed prompts (Xu et al., 2022; Wu & Shi, 2022; Si et al., 2022; Zhu et al., 2023) and demonstrations (Wang et al., 2023b).

## 2.3 ADVERSARIAL DEFENSES

Adversarial defenses fall into two categories: black-box defenses and white-box defenses. In the black-box scenario, we do not have access to model weights (e.g., GPT-4), but we can only observe the outputs of the model. In the white-box scenario, we have full access to the model (e.g., Llama), namely gradient-based defense methods are feasible in this case.

In black-box defenses, defense models are usually treated as pre-processors (Gong et al., 2019; Jones et al., 2020). That is, we first run the defense model to denoise the potentially corrupted input, and then the denoised result is passed to the downstream model for inference.

The white-box defense scheme brings more possibility when designing algorithms since we have full control of the model. However, many existing works define adversarial samples in the continuous embedding space instead of the discrete input space, such that the attacks are not human interpretable (Liu et al., 2020; Cheng et al., 2021; Zuo et al., 2021a;b). Instead, we focus on inputs typos that are plausible in real-world applications.

## 3 SETUPS, ATTACKS, AND DEFENSES

### 3.1 DATASETS, MODELS, AND INFERENCE

**Datasets.** We adopt six datasets from the GLUE benchmark (Wang et al., 2019). For each input sample, we first construct corresponding adversarial samples to simulate input typos. Then, we evaluate LLMs' performance on both the clean data and the corrupted data.

◇ *RTE* (Dagan et al., 2006; Bar-Haim et al., 2006; Giampiccolo et al., 2007; Bentivogli et al., 2009) is a fusion of several annual textual entailment challenges on news articles and headlines. The corresponding task is to determine, given a pair of sentences, whether the meaning of the second one can be inferred from the first one.

⋄ *MRPC* (Dolan & Brockett, 2005) is a set of sentence pairs extracted from news articles. The underlying task is to determine whether two sentences are semantically equivalent.

⋄ *SST-2* (Socher et al., 2013) is a corpus of sentences extracted from online movie reviews, and we need to classify each review as positive or negative.

⋄ *QNLI* (Rajpurkar et al., 2016) is a binary question entailment task. Based on a question-sentence pair, the goal is to determine whether the sentence provides a suitable answer to the question.

⋄ *QQP* (Wang et al., 2019) is a collection of question pairs collected from Quora. The associated task is to determine whether two given questions are paraphrases of each other.

⋄ *MNLI* (Williams et al., 2018) is a crowd-sourced collection of sentence pairs with textual entailment annotations. Given a premise and a hypothesise, the language task is to determine whether the premise entails, contradicts or is neutral with respect to the hypothesis.

**Models.** We adopt two families of Transformer-based (Vaswani et al., 2017) LLMs covering both decoder-only and encoder-decoder architectures.

⋄ *Llama2-Chat* (Touvron et al., 2023b) is a decoder-only model. It is the "chat" version of Llama2 that is trained to follow human instructions (i.e., prompts). We consider two mode sizes: 7B and 13B. Different from the previous version (Touvron et al., 2023a) and many closed-source models, the training of Llama2-Chat emphasizes on model safety.

⋄ *Flan-T5* (Chung et al., 2022) is an encoder-decoder model. We adopt two sizes of Flan-T5: 3B and 11B. We note that Flan-T5 is multi-task instruction fine-tuned on a large number of datasets similar to the natural language understanding tasks we consider.

**Inference.** To evaluate black-box defense methods, we inference LLMs in a zero-shot setting. We list the prompts we adopted to inference Flan-T5 and Llama2-Chat in Appendix A. In the prompts, we instruct the models to choose from a set of possible answers. For example, we instruct the model to choose from *[positive, negative]* for sentiment classification tasks such as SST-2.

## 3.2 ADVERSARIAL TYPOS

We focus on input typos, which is a practical scenario that hinders model performance in real-world applications. In more details, we simulate *worst case input typos* using character-level adversarial attacks. Because our goal is to simulate plausible inputs in actual applications, the corrupted samples should be human interpretable. To facilitate this, for each input sentence, we modify at most 4 words; where in each word, we modify at most one character. The character-level modification can be *insertion*, *deletion* and *substitution*.

---

**Algorithm 1:** Simulate input typos via character-level adversarial attack.

**Input:** Input sample $(x, y)$, where $x$ is the input with $L$ words $x = (x_1, \cdots, x_L)$ and $y$ is the label; Maximum number of words to change $N_{\text{change}}$; Maximum number of tries $N_{\text{try}}$.

**for** $t = 1, \cdots, N_{change}$ **do**

    Randomly select a word $x_n$ from the sentence $x$ that has not been changed yet;

    **for** $i = 1, \cdots, N_{try}$ **do**

        Change $x_n$ to $x_n^i$ by randomly perturbing one character in $x_n$;

        Replace word $x_n$ in sentence $x$ by $x_n^i$, call the corrupted sentence $x^i$;

        Record $p(y|x^i)$, the probability of the ground-truth label $y$;

    Update the sentence $x \leftarrow x^i$, where $i$ is selected that yields the lowest probability $p(y|x^i)$;

**Output:** The corrupted sentence.

---

Algorithm 1 summarizes the attack algorithm we employed. Note that we access the probability of the ground-truth label as a metric to select attacks. For example, in SST-2, the labels are either *positive* or *negative*. Then for a sentence $x$, we retrieve two logits (or logprobs) corresponding to the LLM generating *positive* and *negative*, respectively; after which the logits are normalized to obtain $p(positive|x)$ and $p(negative|x)$. The above procedure is straightforward for open-source LLMs;

and for closed-source LLMs such as GPT-4, publicly available APIs are also provided to access the logprob of the most likely tokens.

Table 1 demonstrates an example of constructing input typos. Initially, the LLM outputs the correct answer with very high confidence. However, after three minor modifications to the text, the LLM makes a wrong prediction with high confidence.

Table 1: An example of constructing a corrupted sample for Flan-T5-3B. *Action* is the action taken in the previous step; *Confidence* is the probability of the ground-truth label (see Algorithm 1).

| Input | Action | Prediction | Label | Confidence |
|---|---|---|---|---|
| that's pure pr hype | — | negative | negative | 1.0 |
| tha**t**'s pure pr hype | deletion | negative | negative | 1.0 |
| tha**t'c**s pure pr hype | insertion | negative | negative | 0.7 |
| tha**t'c**s pure pr hyp**e** | deletion | positive | negative | 0.3 |

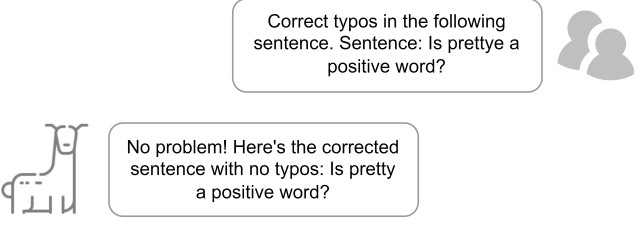

Figure 2: An example of using Llama for typo correction.

Figure 3: An illustration of a rule-based word cluster.

### 3.3 BLACK-BOX DEFENSES

In the black-box scenario, defense methods are usually pre-processors. That is, given a potentially corrupted input, we first run the pre-processor to denoise the input, and the result is subsequently fed to the LLM. We consider several categories of black-box defense methods:

◇ **Rule-based defense.** These methods do not use neural networks for denoising purposes. Instead, they learn mapping rules to convert typos. In this work, we use the most representative rule-based defense: Robust Encoding (Jones et al., 2020). The underlying idea is to systematically assign each word a class representative, and subsequently replace each word in the input by its corresponding representative. An example of a word cluster is shown in Figure 3. Note that in the example, "hello" is the center of the cluster, such that all the other words in the cluster (e.g., "hallo") will be converted to "hello" when running the defense algorithm. Clusters are built based on word frequencies and edit-distance (the number of edits), and exist in two distinct variants.

- *Rule-CC*: Clusters are derived from a connected component clustering on the vocabulary graph, where edges are drawn between edit-distance one words.
- *Rule-Agg*: Clusters are built in an agglomerative manner to balance stability (inclusion of most typos in clusters) and fidelity (consistency within clusters).

It is worth pointing out that there are fewer clusters than words, implying that two real words might be associated to the same representative. Therefore, rule-based defenses may change the meaning of the input (see Table 4 in the experiments as an example).

◇ **Prefix defense.** We modify the prompts in Appendix A by prepending a warning: *In the following question, please pay close attention to typos before answering.* Even though typos will not be directly corrected using this simple strategy, the additional instruction can alert the LLM about potential adversarial contents.

◇ **Neural network (NN) defense.** In this approach, a BART-base (Lewis et al., 2020) model is trained for denoising purposes. BART-base is a sequence-to-sequence Transformer model that contains about 140M parameters, making it much lighter than LLMs. The defense model is trained to recover the initial text (e.g., *pretty*) from the corrupted text (e.g., *prettye*).

◇ **Self defense.** LLMs are extremely powerful and can fulfill most tasks if properly prompted. We investigate whether we can use LLMs themselves to defend against adversarial typos. Figure 2 illustrates that Llama can successfully identify and correct typos. Therefore, in this approach, we inference a LLM twice for a potentially corrupted input. In the first round, we ask the LLM to correct typos, and then we inference the LLM again using the typo-corrected sentence.

### 3.4 WHITE-BOX DEFENSES

The white-box scenario brings more possibility since we have full control of the model. Existing works (Liu et al., 2020; Cheng et al., 2021) rely on the concept of adversarial regularization to improve model robustness. Specifically, adversarial samples in the continuous embedding space are constructed to augment the training data. However, construction of the adversarial samples is extremely slow since multiple forward/backward passes are needed. Such a computational burden is even more severe in the era of LLMs.

We adopt a much simpler yet effective strategy for white-box defense: we fine-tune the LLM on both clean data and corrupted data. The intuition is that once exposed to data with typos, the LLM should recognize the typos' patterns and become robust to them.

## 4 EXPERIMENTS

In all the experiments, for a clean input sentence, we use Algorithm 1 to build a corrupted version of the sentence. We note that the corrupted sentences are model-specific, i.e., the same input sentence will be attacked differently for each model. For black-box defense methods, we first denoise the corrupted sentence by running defense model, and then we feed the denoised output to the LLM to generate the final prediction. For white-box defenses, we directly inference the LLM since it is fine-tuned to be robust to adversarial typos.

### 4.1 ROBUSTNESS OF LLMS

Table 2: Experimental results of Flan-T5 and Llama2-Chat. In the results, *corrupted* means models are evaluated on corrupted data, and *clean* means models are evaluated on clean data without typos.

| | Flan-T5-3B | | Flan-T5-11B | | Llama2-Chat-7B | | Llama2-Chat-13B | |
| --- | --- | --- | --- | --- | --- | --- | --- | --- |
| | clean | corrupted | clean | corrupted | clean | corrupted | clean | corrupted |
| **RTE** | 93.1 | 81.2 | 89.5 | 81.5 | 77.5 | 65.6 | 76.8 | 66.3 |
| **MRPC** | 82.3 | 71.9 | 82.6 | 73.6 | 65.0 | 62.2 | 69.7 | 38.4 |
| **SST-2** | 94.8 | 81.9 | 96.1 | 86.2 | 94.1 | 77.2 | 94.8 | 78.4 |
| **QNLI** | 94.7 | 86.7 | 94.5 | 87.9 | 76.6 | 57.5 | 73.6 | 54.8 |
| **QQP** | 90.9 | 73.4 | 88.7 | 75.8 | 58.6 | 34.9 | 71.4 | 55.6 |
| **MNLI** | 91.8 | 72.9 | 90.3 | 75.4 | 51.1 | 31.0 | 50.5 | 38.9 |
| **Average** | 91.3 | 78.0 | 90.3 | 80.1 | 70.5 | 54.7 | 72.8 | 55.4 |

Table 2 shows experimental results on two versions of Flan-T5 and two versions of Llama2-Chat. We report evaluation results on both the clean data without typos and the corrupted data with typos. From the results, we observe the following regarding model robustness:

◇ LLMs are not robust to adversarial typos. For example, average performance of Flan-T5-11B decreases by 10.2%, and that of Llama2-Chat-13B decreases by 17.4%. The results show that the attack algorithm in Algorithm 1 can successfully identify cases where even small changes to the inputs can drastically change the outputs.

◇ Flan-T5 models are more robust than Llama2-Chat models. For example, average performance of Flan-T5-3B drops by 13.3%, and average performance of Llama2-Chat-7B drops by 15.8% when evaluating on the corrupted data.

We also observe that larger model sizes do not necessarily translate to better performance on all the tasks. For example, on the SST-2 dataset without typos, performance of Llama2-Chat-7B is only

0.7% lower than Llama2-Chat-13B (94.1 vs. 94.8), although the former is about two times smaller. Also, we see that performance of Flan-T5-3B is 91.3% averaged across the six tasks, which is even 1.0% higher than the performance of Flan-T5-11B (90.3%). Such a phenomenon is also observed in existing works (Chung et al., 2022).

## 4.2 BLACK-BOX DEFENSE RESULTS

Table 3: Effectiveness of different black-box defense methods on different models. We report the average performance of the six tasks.

| | No Defense | | Defense Methods | | | | |
|---|---|---|---|---|---|---|---|
| | Clean | Corrupted | Rule-CC | Rule-Agg | Prefix | NN | Self |
| **Flan-T5-3B** | 91.3 | 78.0 | 59.3 | 65.2 | 77.4 | 82.3 | 79.9 |
| **Flan-T5-11B** | 90.3 | 80.1 | 64.4 | 69.6 | 78.6 | 83.3 | 84.1 |
| **Llama2-Chat-7B** | 70.5 | 54.7 | 57.3 | 58.9 | 56.8 | 66.9 | 66.5 |
| **Llama2-Chat-13B** | 72.8 | 55.4 | 51.1 | 54.9 | 57.7 | 67.3 | 69.0 |

Table 4: An example of applying defense methods to a corrupted sentence. Here, ✓ means the model (Llama2-Chat-7B) makes a correct prediction, and ✗ means otherwise.

| Type | Sentence | Correct? |
|---|---|---|
| **Clean** | has all the depth of a wading pool | ✓ |
| **Corrupted** | has all tye depth of a wading pool | ✗ |
| **Rule-CC** | his all the death of a working personal | ✗ |
| **Rule-Agg** | his all the death of a working paul | ✗ |
| **NN** | Has all the depth of a wading pool | ✓ |
| **Self** | It has all the depth of a wading pool | ✓ |

Table 3 demonstrates effectiveness of different black-box defense methods, where we report the average performance of Flan-T5 and Llama2-Chat on the six tasks in Table 2. More details are deferred to Appendix B. We have the following observations:

First, rule-based methods do not work well for defending against adversarial typos. Recall that in rule-based defenses, each word belongs to a specific cluster and the word is converted to its cluster representation. However, because there are much fewer clusters than words, rule-based defense methods can change the meaning of a sentence. We show an example in Table 4 to better understand the rule-based approaches. In the example, "wading pool" in the corrupted sentence (note that these two words are not changed by the adversarial attack) is mapped to "working personal" or "working paul" by rule-based defenses. Such a mapping renders the entire sentence nonsense, and we observe that indeed the LLM makes wrong predictions on the sentence's sentiment. The large model capacities of LLMs enable them to infer the correct words to some extent even without any defense mechanisms. Therefore, after applying rule-based typo correction, when the contextual information loss outweighs the gain brought by typo correction, performance of LLMs drop.

Second, NN-defense and self-defense works well for both Flan-T5 and Llama2-Chat. For example, in Table 3, we see that applying NN-defense brings 3.2% accuracy gain for FlanT5-11B (from 80.1 to 83.3); while applying self-defense brings 4.0% accuracy gain (from 80.1 to 84.1). The performance gain is brought by the fact that both of these defense approaches can successfully correct the typos while maintaining the semantic meaning of the original sentence (see Table 4 for an example).

Third, prefix-defense is effective for LLaMa2-Chat. Recall that in prefix-defense, we prepend an instruction *"In the following question, please pay close attention to typos before answering"* to the prompt. The intuition is that even though typos can not be directly corrected like the other defense methods, the additional instruction can alert the LLM about potential adversarial contents. From the results in Table 3, we see that for Llama2-Chat, model robustness increases by about 2%.

## 4.3 White-Box Defense Results

Existing white-box defense methods leverage adversarial regularization (Cheng et al., 2021; Liu et al., 2020). Specifically, these methods construct continuous adversarial samples in the embedding space to augment the training data. However, such a process is extremely slow since multiple forward and backward passes are needed to construct an adversarial sample. Instead, we propose a simple strategy for white-box defense: we fine-tune LLMs on both clean training data and corrupted training data derived from Algorithm 1.

However, fine-tuning models with billions of parameters can be computationally prohibitive. Therefore, we use an off-the-shelf parameter efficient fine-tuning method: LoRA (Hu et al., 2022). For a weight matrix $W \in \mathbb{R}^{d \times k}$, full fine-tuning computes its gradient with respect to the loss, and updates this weight matrix accordingly. In LoRA, the update of the weight matrix is $W \leftarrow W + BA$, where $B \in \mathbb{R}^{d \times r}$ and $A \in \mathbb{R}^{r \times k}$ are low-rank such that $r \ll \min(d, k)$. We note that the weight matrix $W$ is frozen during fine-tuning, and we only update the two low-rank matrices $A$ and $B$.

In the experiments, we freeze all the weights in the LLM, and we add the LoRA components (i.e., $A$ and $B$) to the query and value matrices in all the attention layers. This is the same strategy as Hu et al. (2022) and has shown to the empirically effective. As a result, for both Flan-T5-3B and Llama2-Chat-7B, we only fine-tune about 5M parameters. We defer training details to Appendix D.

Table 5: Performance of Flan-T5-3B after fine-tuning. We consider two fine-tuning settings: tuning on clean training data only; and tuning on clean and corrupted training data. We evaluate model performance on both the clean development set and the corrupted development set.

| Setting | Test data | RTE | MRPC | SST-2 | QNLI | QQP | MNLI | Average |
|---|---|---|---|---|---|---|---|---|
| **Zero-shot** | clean | 93.1 | 82.3 | 94.8 | 94.7 | 90.9 | 91.8 | 91.3 |
| | corrupted | 81.2 | 71.9 | 81.9 | 86.7 | 73.4 | 72.9 | 78.0 |
| **Fine-tune Clean** | clean | 92.7 | 90.7 | 94.4 | 94.8 | 90.9 | 91.9 | 92.6 |
| | corrupted | 86.5 | 88.0 | 88.7 | 90.9 | 81.0 | 79.9 | 85.8 |
| **Fine-tune Clean + Corr.** | clean | 93.1 | 90.7 | 96.6 | 94.8 | 91.1 | 91.9 | 93.0 |
| | corrupted | 88.0 | 88.8 | 94.0 | 92.6 | 88.6 | 85.0 | 89.5 |

Table 6: Performance of Llama2-Chat-7B after fine-tuning. We consider two fine-tuning settings: tuning on clean training data only; and tuning on clean and corrupted training data. We evaluate model performance on both the clean development set and the corrupted development set.

| Setting | Test data | RTE | MRPC | SST-2 | QNLI | QQP | MNLI | Average |
|---|---|---|---|---|---|---|---|---|
| **Zero-shot** | clean | 77.5 | 65.0 | 94.1 | 76.6 | 58.6 | 51.1 | 70.5 |
| | corrupted | 65.6 | 62.2 | 77.2 | 57.5 | 34.9 | 31.0 | 54.7 |
| **Fine-tune Clean** | clean | 89.5 | 87.9 | 96.2 | 92.6 | 90.9 | 87.8 | 90.8 |
| | corrupted | 85.5 | 86.5 | 89.2 | 89.1 | 86.3 | 83.3 | 86.7 |
| **Fine-tune Clean + Corr.** | clean | 89.5 | 88.4 | 96.4 | 92.6 | 90.8 | 87.8 | 90.9 |
| | corrupted | 87.7 | 87.3 | 93.8 | 90.5 | 89.1 | 85.8 | 89.0 |

Table 5 demonstrates experimental results of Flan-T5-3B; and Table 6 demonstrates experimental results of Llama2-Chat-7B. We consider two fine-tuning schemes: 1) we only fine-tune LLMs on the clean training data; and 2) we fine-tune LLMs on both the clean training data and the corrupted training data that contain typos. From the results, we see that

⋄ Fine-tuning on task-specific clean data improves both model performance and robustness. For example, after fine-tuning, performance of Flan-T5-3B increases by 1.3% on the clean test data and 7.8% on the corrupted test data. As another example, performance of Llama2-Chat-7B increases by 20.3% on the clean data, and robustness increases by 32.0%.

⋄ Fine-tuning on task-specific clean and corrupted data can further improve model robustness. From Table 5, we see that performance of Flan-T5-3B is 89.5% on the corrupted test data, which is more than 20% higher than that in the zero-shot setting.

## 4.4 ATTACKING ADVERSARIAL DEFENSES

Table 3 demonstrates that black-box defenses can indeed improve model robustness. A natural question to ask is: how robust are the defenses against stronger attacks?

Recall that in Algorithm 1, for an input pair $(x, y)$, where $x$ is the input and $y$ is its label, we find an attack $x'$ that empirically minimizes $p(y|x')$. To construct strong attacks, we expose the defense method to the attacker. That is, in the strong attack algorithm, we empirically minimize $p(y|\text{defense}(x'))$, where $\text{defense}(\cdot)$ is the defense method, e.g., a neural typo corrector. In this way, we can build a tailored attack algorithm for each defense approach. Table 7 demonstrates an example of applying strong attacks. We see that with the attack method in Algorithm 1 (i.e., *weak attack*), the NN-defense model can resolve

Table 7: An example of applying NN-defense to corrupted sentences. Here, *weak attack* refers to the result of applying the attack method in Algorithm 1.

| Type | Sentence |
|---|---|
| **Clean** | as vulgar as it is banal |
| **Weak Attack** | a**b** vulgar as **bt s**s banal **c** |
| **NN-defense** | **A** vulgar as **it is** banal **c** |
| **Strong Attack** | a**b** vulgar as **p**it **b**is banal **c** |
| **NN-defense** | **A** vulgar as **p**it **b**is banal **c 18** |

most of the typos. However, when we expose the defense model to the attacker, the resulting attacks are much stronger (i.e., *strong attack*). We see that with strong attacks, the NN-defense model can no longer recover the original semantic meaning.

Table 8: Effectiveness of different black-box defense methods on different models under the strong attack scheme. We report the average performance of the six tasks.

| | Rule-CC | Rule-Agg | Prefix-defense | NN-defense | Self-defense |
|---|---|---|---|---|---|
| **Flan-T5-3B** | 47.3 | 52.0 | 72.4 | 77.1 | 80.1 |
| **Flan-T5-11B** | 50.3 | 55.7 | 71.2 | 77.0 | 82.0 |
| **Llama2-Chat-7B** | 42.8 | 45.4 | 44.7 | 61.2 | 61.3 |
| **Llama2-Chat-13B** | 42.7 | 44.0 | 51.3 | 61.2 | 64.7 |

Table 8 summarizes model performance under strong attacks. Model performance on individual tasks are deferred to Appendix C. We see that the attacks are indeed stronger compared with the weak attacks (see results in Table 3). We also note that even with the strong attacks, performance of NN-defense and self-defense are still satisfactory. For example, performance of the self-defense method on Flan-T5-3B when dealing with the strong attacks is on par with the performance when dealing with the weak attacks.

## 5 CONCLUSION AND DISCUSSION

LLMs have demonstrated superior performance in various natural language processing tasks. However, they are not robust to adversarial attacks. In this work, we consider input typos, which is a practical yet under-explored scenario that hinders performance of LLMs. We systematically study the effectiveness of adversarial defense approaches against input typos. Through extensive experiments, we find that many existing rule-based defense methods that work well on conventional language models are not effective for LLMs. Therefore, we design adversarial defenses tailored for LLMs. First, we develop prompting methods (prefix-defense) to alert the LLM about potential adversarial contents. Second, we find that we can denoise the inputs by adopting either the LLM itself (self-defense) or using a separate small language model (NN-defense). Finally, in the white-box setting, we find that fine-tune LLMs on both clean and corrupted data can be extremely beneficial for model robustness.

In this work, we develop several methods that improve model robustness. Among them, NN-defense is particularly favorable. This is because fine-tuning the LLM may not always be feasible, rendering white-box defenses impractical. Among the black-box defense methods, NN-defense and self-defense are equally effective. However, NN-defense is much faster as only a small language model is used to denoise the input typos, making it more attractive. We note that the proposed prefix-defense is faster than NN-defense and can also improve robustness to some extent. We leave further investigation along this direction as future works.

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

## A    PROMPTS

Below, we show the prompts we used to inference Flan-T5 and Llama2-Chat models.

Table 9: Prompts used to inference Flan-T5.

| Task | Prompt |
|---|---|
| **RTE** | Read the following paragraph and determine if the hypothesis is true: *[input$_1$]*.
Hypothesis: *[input$_2$]*. OPTIONS: yes, no |
| **MRPC** | First sentence: *[input$_1$]*; Second sentence: *[input$_2$]*.
Would you say that these sentences have the same meaning? |
| **SST-2** | Review: *[input$_1$]*. Is this review sentence negative or positive? OPTIONS: positive, negative |
| **QNLI** | Premise: *[input$_1$]*; Hypothesis: *[input$_2$]* Does the premise entail the hypothesis? |
| **QQP** | First question: *[input$_1$]*; Second question: *[input$_2$]*.
Would you say that these questions have the same meaning? |
| **MNLI** | Does the sentence *[input$_1$]* answer the question *[input$_2$]*? |

Table 10: Prompts used to inference Llama2-Chat-7B.

| Task | Prompt |
|---|---|
| **RTE** | *[input$_1$]*. Based on the paragraph above can we conclude that *[input$_2$]*?
Answer with yes or no only. Answer: |
| **MRPC** | Do these sentences mean the same thing? *[input$_1$]*, *[input$_2$]*.
Answer with yes or no only. Answer: |
| **SST-2** | Is the following review positive or negative? Answer in a single word. Review: *[input$_1$]*. Answer: |
| **QNLI** | Does the sentence answer the question? Answer with yes or no only.
Question: *[input$_1$]*. Sentence: *[input$_2$]*. Answer: |
| **QQP** | First sentence: *[input$_1$]*. Second sentence: *[input$_2$]*.
Would you say that these sentences have close meanings? Answer with yes or no only. Answer: |
| **MNLI** | Premise: *[input$_1$]*. Hypothesis: *[input$_2$]*. Is the hypothesis entailed by the premise?
Answer in a single word with yes, no or neutral. Answer: |

Table 11: Prompts used to inference Llama2-Chat-13B.

| Task | Prompt |
|---|---|
| **RTE** | *[input$_1$]*. Based on the paragraph above can we conclude that *[input$_2$]*?
Answer with yes or no only. Answer: |
| **MRPC** | Here are two sentences *[input$_1$]*, *[input$_2$]*.
Do they have the same meaning? Answer with yes or no only. Answer: |
| **SST-2** | Is the following review positive or negative? Answer in a single word. Review: *[input$_1$]*. Answer: |
| **QNLI** | Does the sentence answer the question? Answer with yes or no only.
Question: *[input$_1$]*. Sentence: *[input$_2$]*. Answer: |
| **QQP** | First sentence: *[input$_1$]*. Second sentence: *[input$_2$]*.
Would you say that these sentences have close meanings? Answer with yes or no only. Answer: |
| **MNLI** | Premise: *[input$_1$]*. Hypothesis: *[input$_2$]*. Is the hypothesis entailed by the premise?
Answer in a single word with yes, no or irrelevant. Answer: |

## B    DETAILS OF BLACK-BOX DEFENSE RESULTS

In Table 3, we demonstrate effectiveness of black-box defenses on different models by showing the average score of six tasks. Below, we show detailed breakdown of model performance when applying each defense method on each task.

Table 12: Effectiveness of black-box defense methods on Flan-T5-3B.

| | No Defense | | Defense Methods | | | | |
| | Clean | Corrupted | Rule-CC | Rule-Agg | Prefix | NN | Self |
| --- | --- | --- | --- | --- | --- | --- | --- |
| **RTE** | 93.1 | 81.2 | 59.4 | 65.9 | 81.9 | 80.1 | 82.2 |
| **MRPC** | 82.3 | 71.9 | 41.5 | 47.8 | 65.0 | 68.4 | 74.0 |
| **SST-2** | 94.8 | 81.9 | 70.0 | 77.2 | 83.9 | 91.4 | 86.8 |
| **QNLI** | 94.7 | 86.7 | 66.0 | 71.3 | 87.6 | 88.6 | 87.3 |
| **QQP** | 90.9 | 73.4 | 66.8 | 70.6 | 73.8 | 81.9 | 74.5 |
| **MNLI** | 91.8 | 72.9 | 52.0 | 58.2 | 72.0 | 83.2 | 74.8 |
| **Average** | 91.3 | 78.0 | 59.3 | 65.2 | 77.4 | 82.3 | 79.9 |

Table 13: Effectiveness of black-box defense methods on Flan-T5-11B.

| | No Defense | | Defense Methods | | | | |
| | Clean | Corrupted | Rule-CC | Rule-Agg | Prefix | NN | Self |
| --- | --- | --- | --- | --- | --- | --- | --- |
| **RTE** | 89.5 | 81.5 | 63.8 | 66.7 | 79.7 | 82.2 | 84.1 |
| **MRPC** | 82.6 | 73.6 | 54.6 | 63.4 | 66.5 | 71.6 | 77.5 |
| **SST-2** | 96.1 | 86.2 | 74.3 | 79.8 | 88.4 | 93.3 | 90.8 |
| **QNLI** | 94.5 | 87.9 | 70.1 | 75.9 | 89.6 | 88.0 | 89.4 |
| **QQP** | 88.7 | 75.8 | 69.9 | 71.9 | 71.7 | 81.9 | 81.3 |
| **MNLI** | 90.3 | 75.4 | 53.5 | 60.1 | 75.4 | 82.7 | 81.5 |
| **Average** | 90.3 | 80.1 | 64.4 | 69.6 | 78.6 | 83.3 | 84.1 |

Table 14: Effectiveness of black-box defense methods on Llama2-Chat-7B.

| | No Defense | | Defense Methods | | | | |
| | Clean | Corrupted | Rule-CC | Rule-Agg | Prefix | NN | Self |
| --- | --- | --- | --- | --- | --- | --- | --- |
| **RTE** | 77.5 | 65.6 | 60.1 | 61.2 | 66.3 | 73.9 | 72.5 |
| **MRPC** | 65.0 | 62.2 | 59.8 | 59.7 | 65.7 | 65.5 | 65.7 |
| **SST-2** | 94.1 | 77.2 | 67.3 | 71.2 | 80.3 | 89.8 | 88.1 |
| **QNLI** | 76.6 | 57.5 | 56.9 | 59.9 | 54.7 | 64.4 | 69.2 |
| **QQP** | 58.6 | 34.9 | 61.0 | 60.1 | 45.4 | 59.2 | 56.2 |
| **MNLI** | 51.1 | 31.0 | 38.4 | 41.2 | 28.4 | 48.6 | 47.4 |
| **Average** | 70.5 | 54.7 | 57.3 | 58.9 | 56.8 | 66.9 | 66.5 |

Table 15: Effectiveness of black-box defense methods on Llama2-Chat-13B.

| | No Defense | | Defense Methods | | | | |
| | Clean | Corrupted | Rule-CC | Rule-Agg | Prefix | NN | Self |
| --- | --- | --- | --- | --- | --- | --- | --- |
| **RTE** | 76.8 | 66.3 | 51.1 | 55.1 | 58.7 | 71.0 | 69.9 |
| **MRPC** | 69.7 | 38.4 | 38.7 | 45.5 | 65.9 | 62.9 | 66.6 |
| **SST-2** | 94.8 | 78.4 | 61.4 | 68.7 | 74.2 | 88.5 | 90.7 |
| **QNLI** | 73.6 | 54.8 | 51.8 | 53.3 | 52.2 | 63.6 | 69.1 |
| **QQP** | 71.4 | 55.6 | 63.9 | 64.2 | 60.6 | 70.3 | 70.2 |
| **MNLI** | 50.5 | 38.9 | 39.9 | 42.6 | 34.3 | 47.5 | 47.5 |
| **Average** | 72.8 | 55.4 | 51.1 | 54.9 | 57.7 | 67.3 | 69.0 |

# C    DETAILS OF DEFENSE AGAINST STRONG ATTACKS

In Table 8, we demonstrate effectiveness of black-box defenses against strong attacks. Specifically, for each model, we show the average score of six tasks. Below, we show detailed breakdown of model performance when applying each defense method on each task.

Table 16: Effectiveness of black-box defenses against strong attacks on Flan-T5-3B.

|  | No Defense | | Defense Methods | | | | |
|---|---|---|---|---|---|---|---|
|  | **Clean** | **Corrupted** | **Rule-CC** | **Rule-Agg** | **Self** | **Prefix** | **NN** |
| **RTE** | 93.1 | 81.2 | 46.7 | 50.4 | 81.5 | 76.8 | 79.0 |
| **MRPC** | 82.3 | 71.9 | 35.7 | 38.6 | 74.8 | 58.2 | 62.5 |
| **SST-2** | 94.8 | 81.9 | 55.3 | 61.7 | 85.0 | 81.3 | 85.1 |
| **QNLI** | 94.7 | 86.7 | 54.5 | 60.4 | 88.2 | 86.6 | 84.4 |
| **QQP** | 90.9 | 73.4 | 58.6 | 59.8 | 75.7 | 66.8 | 77.1 |
| **MNLI** | 91.8 | 72.9 | 33.1 | 41.1 | 75.1 | 64.6 | 74.2 |
| **Average** | 91.3 | 78.0 | 47.3 | 52.0 | 80.1 | 72.4 | 77.1 |

Table 17: Effectiveness of black-box defenses against strong attacks on Flan-T5-11B.

|  | No Defense | | Defense Methods | | | | |
|---|---|---|---|---|---|---|---|
|  | **Clean** | **Corrupted** | **Rule-CC** | **Rule-Agg** | **Self** | **Prefix** | **NN** |
| **RTE** | 89.5 | 81.5 | 54.0 | 56.5 | 82.2 | 72.1 | 76.1 |
| **MRPC** | 82.6 | 73.6 | 40.3 | 47.7 | 70.0 | 54.2 | 63.5 |
| **SST-2** | 96.1 | 86.2 | 55.6 | 64.9 | 90.6 | 82.8 | 86.6 |
| **QNLI** | 94.5 | 87.9 | 55.9 | 63.5 | 89.7 | 86.5 | 84.9 |
| **QQP** | 88.7 | 75.8 | 57.7 | 58.5 | 81.5 | 63.1 | 75.8 |
| **MNLI** | 90.3 | 75.4 | 38.4 | 43.0 | 78.2 | 68.7 | 75.3 |
| **Average** | 90.3 | 80.1 | 50.3 | 55.7 | 82.0 | 71.2 | 77.0 |

Table 18: Effectiveness of black-box defenses against strong attacks on Llama2-Chat-7B.

|  | No Defense | | Defense Methods | | | | |
|---|---|---|---|---|---|---|---|
|  | **Clean** | **Corrupted** | **Rule-CC** | **Rule-Agg** | **Self** | **Prefix** | **NN** |
| **RTE** | 77.5 | 65.6 | 40.9 | 46.4 | 64.5 | 53.3 | 68.8 |
| **MRPC** | 65.0 | 62.2 | 48.7 | 51.2 | 65.1 | 65.7 | 64.4 |
| **SST-2** | 94.1 | 77.2 | 49.4 | 57.1 | 85.2 | 67.0 | 84.8 |
| **QNLI** | 76.6 | 57.5 | 45.7 | 47.4 | 63.0 | 43.4 | 61.0 |
| **QQP** | 58.6 | 34.9 | 49.3 | 45.4 | 47.9 | 22.8 | 46.4 |
| **MNLI** | 51.1 | 31.0 | 22.8 | 25.1 | 42.3 | 16.1 | 41.7 |
| **Average** | 70.5 | 54.7 | 42.8 | 45.4 | 61.3 | 44.7 | 61.2 |

# D    MORE DETAILS ABOUT WHITE-BOX DEFENSE

We implement white-box defenses using PyTorch (Paszke et al., 2019) and the HuggingFace code-base (Wolf et al., 2019). In all the experiments, we use a batch size of 64 and we use AdamW (Loshchilov & Hutter, 2019) as the optimizer. The other hyper-parameters are shown in Table 20 and Table 21.

Table 19: Effectiveness of black-box defenses against strong attacks on Llama2-Chat-13B.

| | No Defense | | Defense Methods | | | | |
| | Clean | Corrupted | Rule-CC | Rule-Agg | Self | Prefix | NN |
|---|---|---|---|---|---|---|---|
| **RTE** | 76.8 | 66.3 | 44.2 | 48.2 | 63.8 | 53.6 | 68.8 |
| **MRPC** | 69.7 | 38.4 | 33.7 | 32.3 | 62.2 | 65.1 | 52.5 |
| **SST-2** | 94.8 | 78.4 | 47.9 | 53.0 | 89.2 | 62.1 | 81.5 |
| **QNLI** | 73.6 | 54.8 | 49.1 | 48.0 | 62.2 | 46.7 | 57.3 |
| **QQP** | 71.4 | 55.6 | 56.0 | 53.7 | 67.2 | 46.5 | 63.4 |
| **MNLI** | 50.5 | 38.9 | 25.3 | 28.5 | 43.3 | 33.6 | 43.5 |
| **Average** | 72.8 | 55.4 | 42.7 | 44.0 | 64.7 | 51.3 | 61.2 |

Table 20: Hyer-parameters for fine-tuning Flan-T5-3B. Here, *LR* is the learning rate, and *LoRA-alpha* is an initialization parameter in Hu et al. (2022). We consider two settings: *clean* means we only train on clean training data, and *c+c* means we train on clean and corrupted training data.

| | LR | | Epoch | | Warmup | | LoRA-rank | | LoRA-alpha | |
| | clean | c+c | clean | c+c | clean | c+c | clean | c+c | clean | c+c |
|---|---|---|---|---|---|---|---|---|---|---|
| **RTE** | 4e-4 | 3e-4 | 30 | 30 | 0.1 | 0.2 | 32 | 8 | 64 | 16 |
| **MRPC** | 1e-4 | 3e-4 | 20 | 10 | 0.1 | 0.1 | 8 | 8 | 16 | 16 |
| **SST-2** | 5e-4 | 5e-4 | 10 | 10 | 0 | 0.1 | 8 | 8 | 16 | 16 |
| **QNLI** | 1e-3 | 1e-3 | 10 | 10 | 0.1 | 0.2 | 8 | 8 | 16 | 16 |
| **QQP** | 1e-3 | 7e-4 | 20 | 10 | 0.1 | 0.1 | 8 | 8 | 16 | 16 |
| **MNLI** | 7e-4 | 3e-4 | 10 | 10 | 0.1 | 0.1 | 8 | 8 | 16 | 16 |

Table 21: Hyer-parameters for fine-tuning Llama2-Chat-7B. Here, *LR* is the learning rate, and *LoRA-alpha* is an initialization parameter in Hu et al. (2022). We consider two settings: *clean* means we only train on clean training data, and *c+c* means we train on clean and corrupted training data.

| | LR | | Epoch | | Warmup | | LoRA-rank | | LoRA-alpha | |
| | clean | c+c | clean | c+c | clean | c+c | clean | c+c | clean | c+c |
|---|---|---|---|---|---|---|---|---|---|---|
| **RTE** | 2e-4 | 2e-4 | 30 | 30 | 0.1 | 0.1 | 32 | 32 | 64 | 64 |
| **MRPC** | 7e-4 | 7e-4 | 20 | 20 | 0.1 | 0.1 | 8 | 8 | 16 | 16 |
| **SST-2** | 5e-4 | 5e-4 | 10 | 10 | 0.1 | 0.1 | 8 | 8 | 16 | 16 |
| **QNLI** | 7e-4 | 2e-4 | 10 | 10 | 0.1 | 0.1 | 8 | 8 | 16 | 16 |
| **QQP** | 1e-3 | 7e-4 | 10 | 10 | 0.1 | 0.1 | 8 | 8 | 16 | 16 |
| **MNLI** | 5e-4 | 5e-4 | 10 | 10 | 0.1 | 0.1 | 8 | 8 | 16 | 16 |

