# OpenReview forum: "Evaluating Adversarial Defense in the Era of Large Language Models"
_ICLR.cc/2024/Conference — ICLR 2024 Conference Withdrawn Submission_

### Official Review · Reviewer_GNz2 · 2023-10-30

**Soundness:** 2 fair
**Presentation:** 2 fair
**Contribution:** 2 fair
**Rating:** 3
**Confidence:** 4

**Summary:**

In this work, the authors examine LLM robustness to adversarial typos in user prompts. They propose an algorithm for finding typo-style attacks that maximally reduce model performance and they propose several defense tactics to mitigate the harm of this type of attack. Of the defenses considered, five operate without explicit knowledge of the LLM weights -- a setting the authors call Black-Box, and one includes fine tuning the LLM and therefore can only be done when the weights are accessible, or in the White-Box setting. The authors establish that typos can reduce performance and they show with empirical results on a handful of datasets spanning several downstream tasks that some of the defenses proposed can help mitigate that harm.

**Strengths:**

This paper has several strengths.
1. It considers a range of datasets/tasks -- providing experimental results from six down stream tasks
2. A range of attacks are proposed and analyzed -- one weak and one strong attack.
3. A wide variety of defenses are discussed and used in experimentation.
4. The exposition is clear -- well written and organized.

**Weaknesses:**

1. The attack is constrained to be (a series of) single character perturbations -- in otherwords standard looking typos, but the average spellcheckers are not mentioned. I understand that open source models like Llama don't include preprocessing and that we largely don't know what proprietary systems like ChatGPT do with user input (in the way of preprocessing). But in the vein of the attacks and defenses considered in this work I think a standard spell-check or grammar editor is a reasonable baseline (see more in the questions below).

2. The threat model is confusing to me in this context. I think the authors defined the terms "white-box" and "black-box" and use them consistently, but what setting would a defense be needed where the defender knows the type of attack, but does not have access to the weights of the model? (Better phrasing in the questions below.) Overall, I think the threat model is contrived.

3. The conclusions are foggy. I understand that in works that consider a range of methods toward the same end -- in this case defense against typos -- that sometimes the conclusion is that some methods work in some situations and others work better in other situations. But I didn't draw the same conclusion as the authors describe in the Conclusion and Discussion section. Firstly, they seem to write off white-box defenses as impractical, and while this claim has merit it seems to odd to introduce the white box method after the black box only to back pedal and say these aren't practical. Also, I'd like some clarity on the threat model so I can better understand when the black box methods are, in fact more practical.

4. The attack may actually change the labels. The authors don't adequately address how often the attack changes the semantic meaning of the input. This is a critical step in this project since some of the downstream tasks are related to semantic meaning. Historically, adversarial robustness work discusses "clean label" attacks or defines "adversarial" as imperceptible to humans or as something that doesn't change the semantic meaning, and enforces this constraint with some norm (l-inf is common in vision). In this work, how do we know the perturbations aren't changing the semantics?

I don't think the following weakness keeps me from accepting this work, but it is the cause of my **Presentation** score above, which asks to consider the contextual place among related work. I do think this is a weakness and I think the authors should consider addressing some of the more recent work on LLM robustness in the related work section. The particular related work referenced below is only a small subset of existing work on LLM robustness and it does generally consider a slightly different adversarial objective -- but I still feel that without mentioning the work from the past summer this paper is lacking.

1. This paper feels out-of-date. The details in the reviewer guide say that work published on or after May 28, 2023 does not need to be compared to but the authors may reference or discuss it. In this instance -- in this outrageously fast paced field I think we have a particularly difficult situation. The title of this work "Evaluating Adversarial Defense in the Era of Large Language Models" and the work on the topic that came out over this past summer make it feel to me like it is quite behind the times. To be clear, I'm not critical of the authors -- I routinely find my own work in the same category -- but as a reviewer, I am hesitant.
    1. [Universal and Transferable Adversarial Attacks
on Aligned Language Models by Zhou et al. 2023](https://arxiv.org/pdf/2307.15043.pdf?utm_referrer=https%3A%2F%2Fdzen.ru%2Fmedia%2Fid%2F64b465f36e39c84dadcc5f47%2F64cccb22226a2d25f10e8379) Presents a far stronger attack than typo-style manipulations and it has been cited 45 times (according to Google Scholar at the time of writing this review).
    2. Follow up work on that attack method that proposes defenses to it has also come out.
          1. [Baseline Defenses for Adversarial Attacks Against Aligned Language Models by Jain et al. 2023](https://arxiv.org/pdf/2309.00614.pdf) has been cited 14 times (according to Semantic Scholar at the time of writing this review). This work looks similar to the work at the focus of this review as it highlights several defenses and adaptive attacks (like the 'strong' attack).
          2. [Certifying LLM Safety Against Adversarial Prompting by Kumar et al. 2023](https://arxiv.org/pdf/2309.02705) proposes certifiable defenses and has also been cited already.

**Questions:**

1. Spell checker -- How does something like [pyspellchecker](https://pypi.org/project/pyspellchecker/) or [spellwise](https://github.com/chinnichaitanya/spellwise) do? I think if the attack is supposed to look like reasonable typos, fast existing spell checkers should be used on every word as a baseline. These are just two Python packages I found, but any pipeline for addressing typos (without an LLM) would suffice here as a baseline. Can these attacks be removed with a process like these?
2. Can the authors motivate the threat model better?
    1. What is a setting where the defender would know that the attack might look like typos (an assumption for black-box and white-box defenses considered in this work)?
    2. When do we generally encounter a situation where a defender is accepting input that might be attacked and they need to return LLM output, but they can't access the LLM weights (an assumption for black box methods)?
3. How often would the attacked inputs no longer get the same label as the clean inputs? This feels critical to see in order to contextualize the accuracy numbers in all the tables.


I look forward to engaging with the authors and the other reviewers and I'm generally open to increasing my score if my concerns are addressed.

---

### Official Review · Reviewer_WaJ4 · 2023-11-03

**Soundness:** 2 fair
**Presentation:** 2 fair
**Contribution:** 2 fair
**Rating:** 3
**Confidence:** 3

**Summary:**

In this paper, the authors discuss the robustness of LLMs against adversarial attacks and propose defense mechanisms against such attacks. Specifically, the authors focus on the real-world scenario of typing errors and validate the model's vulnerability using Llama2-Chat and Flan-T5. Additionally, the authors introduce both white-box and black-box strategies for their approach.


Overall, I think this paper discusses an important issue, but it lacks novelty and is limited. I greatly encourage the authors to further this research and consider more complex attack scenarios.

**Strengths:**

1. This paper investigates an important topic the robustness of LLM, and experimentally verifies the vulnerability of LLM to adversarial attacks.

2. This paper is well-written and easy to understand.

**Weaknesses:**

1. While typing errors are prevalent in real-life, they represent a limited form of adversarial attack with limited impact. As a result, the transferability and generalizability of the proposed defenses are poor.

2. The key concern about the paper is the lack of novelty. Novelty appears limited to combining existing approaches, the authors should more clearly describe their specific novel contributions.

3. In the white-box scenario, direct adversarial training of LLMs is costly and suffers from catastrophic forgetting. The cost of adversarial training is also unclear.

**Questions:**

see the above.

---

### Official Review · Reviewer_M5vg · 2023-11-03

**Soundness:** 2 fair
**Presentation:** 2 fair
**Contribution:** 1 poor
**Rating:** 3
**Confidence:** 4

**Summary:**

The authors propose different black and white box defences against adversarial attacks which aim to generate typos in the input text. Firstly the typo attack is generated by modifying at most four words. On testing the corrupted input using a standard trained model without any test time modification of the input sample, the authors observe over 10-17% drop in accuracy indicating that the current LLMs like Flan and Llama are not robust to typo attacks.  The authors then demonstrate that rule based defences are also not able to improve the robustness to a large extent. Motivated by this, the authors propose different ways to achieve robustness in black and white box settings. In the black box setting, the authors propose to append a prefix before the text, train a smaller neural network to output text without typos, use self defense by appending a prefix before the text and asking the model itself to generate the text without typos. In the white box setting, the authors propose to fine-tune the model on corrupted as well as clean data. Results demonstrate that the proposed approaches are effective in defending the model against typos in text.

**Strengths:**

The ideas presented by the authors are simple and the evaluation is done on models with different number of parameters to develop a better understanding of the proposed defences.

**Weaknesses:**

Major Weaknesses:

* The robustness of the proposed defences is not evaluated properly. It seems likely that an adaptive attack can fool the model. For instance, in the case of using a smaller model to correct the text with wrong spelling, the attacker could try to generate an attack such that the original black box model as well as the smaller model can be fooled simultaneously. Similarly in the case of self defense, the attacker can try to fool the model by appending the text with the appended prompt. Therefore, the authors should consider a more realistic scenario and then try to evaluate their defence by considering these conditions.

* Similarly in the case of white box attacks, there is no proper justification to believe why crafting an adaptive attack is not possible. It has already been shown that fine-tuning doesn't change the mechanisms of the model, though it might change the model's behavior[1]. As a result it is unexpected that fine-tuning would make the model robust. This robustness might be seen only on the fine tuning distribution of samples and against weak attacks.

Minor weaknesses:
* I feel that the scope of the problem addressed is a bit narrow and the authors should try to investigate a more general version of adversarial attacks rather than only investigating the attacks which aim to misspell the sentences.

* It would be nice if the authors try to focus on one defence method and provide a comprehensive analysis on it, instead of proposing multiple defences. This will help in improving the focus of the paper.

[1] Qi, Xiangyu et al. “Fine-tuning Aligned Language Models Compromises Safety, Even When Users Do Not Intend To!” ArXiv abs/2310.03693 (2023)

**Questions:**

I would request the authors to kindly address the comments in the weakness section.

---

### Official Review · Reviewer_VoDX · 2023-11-05

**Soundness:** 3 good
**Presentation:** 3 good
**Contribution:** 2 fair
**Rating:** 3
**Confidence:** 3

**Summary:**

This submission systematically evaluates the LLM robustness against character-level typo-like attacks. The benchmark considers a few classification tasks in the NLP domain, Llama2-Chat and Flan-T5 models with parameters between 3B and 13B, and several black-box and white-box defenses. Results show that adversarial typos significantly hurt model performance. As a conclusion, among black-box defenses, denoising the inputs with the model itself or a separate smaller is the most effective one; among white-box defenses, fine-tuning on both clean and corrupted data is the most effective, and white-box defenses are much stronger than black-box ones.

**Strengths:**

- A comprehensive study of LLM robustness against adversarial typos, bringing a few interesting findings and takeaways for practitioners.

- The writing quality is generally good and it is easy to follow.

**Weaknesses:**

- I feel the overall contribution might be limited. The safety threat of character-level word substitution has been found and studied long before. The defense techniques are also not novel. In light of this, the submission is like expanding the study breadth in LLMs. Moreover, the study is particularly focused on character substitution attacks. In natural language domain, the robustness threat comes from much broader aspects. Even for input perturbation attacks, character substitution is just a specific type. There are several others like word substitution, word removal, word insertion, sentence paraphrasing, etc, c.f., AdvGLUE. The conclusions from one threat model may not be able to improve our standing on the general topic of robustness in LLMs.

**Questions:**

For rule-based defenses, is it possible to make them more effective by increasing the number of clusters? If we make the clusters more than words, then two real words might not be compressed to the same representative and the rule-based defenses may not change the meaning of the inputs, so these defenses can be more effective.

**Details Of Ethics Concerns:**

The submission may need a separate statement on how the robustness evaluation implies the general robustness rank or degree of LLMs by highlighting its scope and limitations to prevent results from being overly interpreted.